# OPERATOR LEARNING MEETS NUMERICAL ANALYSIS: IMPROVING NEURAL NETWORKS THROUGH ITERATIVE METHODS

## ABSTRACT

Deep neural networks, despite their success in numerous applications, often function without established theoretical foundations. In this paper, we bridge this gap by drawing parallels between deep learning and classical numerical analysis. By framing neural networks as operators with fixed points representing desired solutions, we develop a theoretical framework grounded in iterative methods for operator equations. Under defined conditions, we present convergence proofs based on fixed point theory. We demonstrate that popular architectures, such as diffusion models and AlphaFold, inherently employ iterative operator learning. Empirical assessments highlight that performing iterations through network operators improves performance. We also introduce an iterative graph neural network, PIGN, that further demonstrates benefits of iterations. Our work aims to enhance the understanding of deep learning by merging insights from numerical analysis, potentially guiding the design of future networks with clearer theoretical underpinnings and improved performance.

## 1 INTRODUCTION

Deep neural networks have become essential tools in domains such as computer vision, natural language processing, and physical system simulations, consistently delivering impressive empirical results. However, a deeper theoretical understanding of these networks remains an open challenge. This study seeks to bridge this gap by examining the connections between deep learning and classical numerical analysis.

By interpreting neural networks as operators that transform input functions to output functions, discretized on some grid, we establish parallels with numerical methods designed for operator equations. This approach facilitates a new iterative learning framework for neural networks, inspired by established techniques like the Picard iteration.

Our findings indicate that certain prominent architectures, including diffusion models, AlphaFold, and Graph Neural Networks (GNNs), inherently utilize iterative operator learning (see Figure 1). Empirical evaluations show that adopting a more explicit iterative approach in these models can enhance performance. Building on this, we introduce the Picard Iterative Graph Neural Network (PIGN), an iterative GNN model, demonstrating its effectiveness in node classification tasks.

In summary, our work:

- Explores the relationship between deep learning and numerical analysis from an operator perspective.
- Introduces an iterative learning framework for neural networks, supported by theoretical convergence proofs.
- Evaluates the advantages of explicit iterations in widely-used models.
- Presents PIGN and its performance metrics in relevant tasks.
- Provides insights that may inform the design of future neural networks with a stronger theoretical foundation.

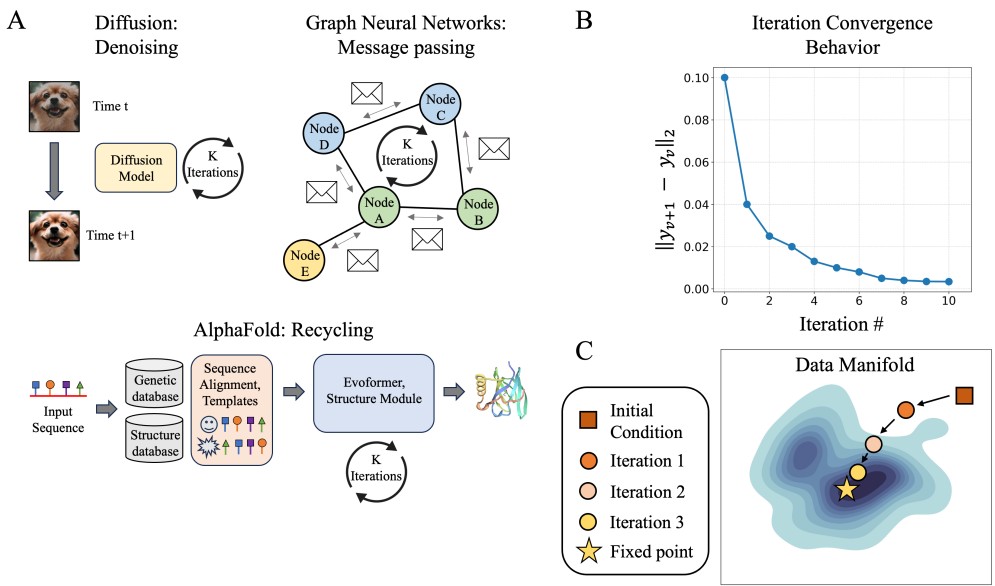

Figure 1: Overview of iterative framework. (A) Popular architectures which incorporate iterative components in their framework. (B) Convergence behavior of an iterative solver. (C) Behavior of iterative solver converging to a fixed point in the data manifold.

The remainder of this manuscript is organized as follows: We begin by delving into the background and related work to provide the foundational understanding for our contributions. This is followed by an introduction to our theoretical framework for neural operator learning. Subsequently, we delve into a theoretical exploration of how various prominent deep learning frameworks undertake operator learning. We conclude with empirical results underscoring the advantages of our proposed framework.

## 2 BACKGROUND AND RELATED WORK

**Numerical Analysis.** Numerical analysis is rich with algorithms designed for approximating solutions to mathematical problems. Among these, the Banach-Caccioppoli theorem is notable, used for iteratively solving operator equations in Banach spaces. The iterations, often called Fixed Point iterations, or Picard iterations, allow to solve an operator equation approximately, in an iterative manner. Given an operator $T$, this approach seeks a function $u$ such that $T(u) = u$, called a fixed point, starting with an initial guess and refining it iteratively.

The use of iterative methods has a long history in numerical analysis for approximate solutions of intractable equations, for instance involving nonlinear operators. For example, integral equations, e.g. of Urysohn and Hammerstein type, arise frequently in physics and engineering applications and their study has long been treated as a fixed point problem (Krasnosel'skii, 1964; Atkinson & Han, 2005; Atkinson & Potra, 1987).

Convergence to fixed points can be guaranteed under contractivity assumptions by the Banach-Caccioppoli fixed point theorem (Atkinson & Han, 2005). Iterative solvers have also been crucial for partial differential equations and many other operator equations (Kelley, 1995).

**Operator Learning.** Operator learning is a class of deep learning methods where the objective of optimization is to learn an operator between function spaces. Examples and an extended literature can be found in Kovachki et al. (2021); Lu et al. (2021). The interest of such an approach, is that mapping functions to functions we can model dynamics datasets, and leverage the theory of operators. When the operator learned is defined through an equation, e.g. an integral equation as in Zappala et al. (2022), along with the training procedure we also need a way of solving said equation, i.e. we need a solver. For highly nonlinear problems, when deep learning is not involved,

these solvers often utilize some iterative procedure as in Kelley (1995). Our approach here brings the generality of iterative approaches into deep learning by allowing to learn operators between function spaces through iterative procedure used in solving nonlinear operator equations.

**Transformers.** Transformers (Vaswani et al. (2017), Devlin et al. (2019), Radford & Narasimhan (2018)), originally proposed for natural language processing tasks, have recently achieved state-of-the-art results in a variety of computer vision applications (Dosovitskiy et al. (2020), Chen et al. (2020), He et al. (2022), Baevski et al. (2022), Assran et al. (2023)). Their self-attention mechanisms make them well-suited for tasks beyond just sequence modeling. Notably, transformers have been applied in an iterative manner in some contexts, such as the "recycling" technique used in AlphaFold2 (Jumper et al., 2021).

**AlphaFold.** DeepMind's AlphaFold (Senior et al., 2020) is a protein structure prediction model, which was significantly improved in Jumper et al. (2021) with the introduction of AlphaFold2 and further extended to protein complex modeling in AlphaFold-Multimer (Evans et al., 2022). AlphaFold2 employs an iterative refinement technique called "recycling", which recycles the predicted structure through its entire network. The number of iterations was increased from 3 to 20 in AF2Complex (Gao et al., 2022), where improvement was observed. An analysis of DockQ scores with increased iterations can be found in Johansson-Åkhe & Wallner (2022). We only look at monomer targets, where DockQ scores do not apply and focus on global distance test (GDT) scores and root-mean-square deviation (RMSD).

**Diffusion Models.** Diffusion models were first introduced in Sohl-Dickstein et al. (2015) and were shown to have strong generative capabilities in Song & Ermon (2019) and Ho et al. (2020). They are motivated by diffusion processes in non-equilibrium thermodynamics (Jarzynski, 1997) related to Langevin dynamics and the corresponding Kolmogorov forward and backward equations. Their connection to stochastic differential equations and numerical solvers is highlighted in Song et al. (2021), Kingma et al. (2021), Dockhorn et al. (2022), and Meng et al. (2022). We focus on the performance of diffusion models at different amounts of timesteps used during training, including an analysis of FID (Heusel et al., 2017) scores.

**Graph Neural Networks (GNNs).** GNNs are designed to process graph-structured data through iterative mechanisms. Through a process called message passing, they repeatedly aggregate and update node information, refining their representations. The iterative nature of GNNs was explored in Tang et al. (2020), where the method combined repeated applications of the same GNN layer using confidence scores. Although this shares similarities with iterative techniques, our method distinctly leverages fixed-point theory, offering specific guarantees and enhanced performance, as detailed in Section 5.1.

## 3 ITERATIVE METHODS FOR SOLVING OPERATOR EQUATIONS

In the realm of deep learning and neural network models, direct solutions to operator equations often become computationally intractable. This section offers a perspective that is applicable to machine learning, emphasizing the promise of iterative methods for addressing such challenges in operator learning. We particularly focus on how the iterative numerical methods converge and their application to neural network operator learning. These results will be used in the Appendix to derive theoretical convergence guarantees for iterations on GNNs and Transformer architectures, see Appendix A.

### 3.1 SETTING AND PROBLEM STATEMENT

Consider a Banach space $X$. Let $T : X \longrightarrow X$ be a continuous operator. Our goal is to find solutions to the following equation:

$$\lambda T(x) + f = x, \tag{1}$$

where $f \in X$ and $\lambda \in \mathbb{R} - \{0\}$ is a nontrivial scalar. A solution to this equation is a fixed point $x^*$ for the operator $P = \lambda T + f$:

$$\lambda T(x^*) + f = x^*. \tag{2}$$

## 3.2 ITERATIVE TECHNIQUES

It is clear that for arbitrary nonlinear operators, solving Equation (1) is not feasible. Iterative techniques such as Picard or Newton-Kantorovich iterations become pivotal. These iterations utilize a function $g$ and progress as:

$$x_{n+1} = g(T, x_n). \tag{3}$$

Central to our discussion is the interplay between iterative techniques and neural network operator learning. We highlight the major contribution of this work: By using network operators iteratively during training, convergence to network fixed points can be ensured. This approach uniquely relates deep learning with classical numerical techniques.

## 3.3 CONVERGENCE OF ITERATIONS AND THEIR APPLICATION

A particular case of great interest is when the operator $T$ takes an integral form and $X$ represents a function space, our framework captures the essence of an integral equation (IE). By introducing $P_\lambda(x) = \lambda T(x) + f$, we can rephrase our problem as a search for fixed points.

We now consider the problem of approximating a fixed point of a nonlinear operator. The results of this section are applied to various deep learning settings in Appendix A to obtain theoretical guarantees for the iterative approaches.

**Theorem 1.** *Let $\epsilon > 0$ be fixed, and suppose that $T$ is Lipschitz with constant $k$. Then, for all $\lambda$ such that $|\lambda k| < 1$, we can find $y \in X$ such that $||\lambda T(y) + f - y|| < \epsilon$ for any choice of $\lambda$, independently of the choice of $f$.*

*Proof.* Let us set $y_0 := f$ and $y_{n+1} = f + \lambda T(y_n)$ and consider the term $||y_1 - y_0||$. We have

$$||y_1 - y_0|| = ||\lambda T(y_0)|| = |\lambda| ||T(y_0)||.$$

For an arbitrary $n > 1$ we have

$$||y_{n+1} - y_n|| = ||\lambda T(y_n) - \lambda T(y_{n-1})|| \le k|\lambda| ||y_n - y_{n-1}||.$$

Therefore, applying the same procedure to $y_n - y_{n-1} = T(y_{n-1}) - T(y_{n-2})$ until we reach $y_1 - y_0$, we obtain the inequality

$$||y_{n+1} - y_n|| \le |\lambda|^n k^n ||T(y_0)||.$$

Since $|\lambda| k < 1$, the term $|\lambda|^n k^n ||T(y_0)||$ is eventually smaller than $\epsilon$, for all $n \ge \nu$ for some choice of $\nu$. Defining $y := y_\nu$ for such $\nu$ gives the following

$$||\lambda T(y_\nu) + f - y_\nu|| = ||y_{\nu+1} - y_\nu|| < \epsilon.$$

$\square$

The following now follows easily.

**Corollary 1.** *Consider the same hypotheses as above. Then Equation 1 admits a solution for any choice of $\lambda$ such that $|\lambda| k < 1$.*

*Proof.* From the proof of Theorem 1 it follows that the sequence $y_n$ is a Cauchy sequence. Since $X$ is Banach, then $y_n$ converges to $y \in X$. By continuity of $T$, $y$ is a solution to Equation 1. $\square$

Recall that for nonlinear operators, continuity and boundedness are not equivalent conditions.

**Corollary 2.** *If in the same situation above $T$ is also bounded, then the choice of $\nu$ of the iteration can be chosen uniformly with respect to $f$, for a fixed choice of $\lambda$.*

*Proof.* From the proof of Theorem 1, we have that

$$||y_{n+1} - y_n|| \le |\lambda|^n k^n ||T(y_0)|| = |\lambda|^n k^n ||T(f)||.$$

If $T$ is bounded by $M$, then the previous inequality is independent of the element $f \in X$. Let us choose $\nu$ such that $|\lambda|^n k^n < \epsilon/M$. Then, suppose $f$ is an arbitrary element of $X$. Initializing $y_0 = f$, $y_\nu$ will satisfy $||\lambda T(y_\nu) + f - y_\nu|| < \epsilon$, for any given choice of $\epsilon$. $\square$

The following result is classic, and its proof can be found in several sources. See for instance Chapter 5 in Atkinson & Han (2005).

**Theorem 2.** *(Banach-Caccioppoli fixed point theorem) Let $X$ be a Banach space, and let $T : X \longrightarrow X$ be contractive mapping with contractivity constant $0 < k < 1$. Then, $T$ has a unique fixed point, i.e. the equation $T(x) = x$ has a unique solution $u$ in $X$. Moreover, for any choice of $u_0$, $u_n = T^n(u_0)$ converges to the solution with rate of convergence*

$$||u_n - u|| \quad < \quad \frac{k^n}{1-k}||u_0 - u_1||, \tag{4}$$

$$||u_n - u|| \quad < \quad \frac{k}{1-k}||u_{n-1} - u_n||. \tag{5}$$

The possibility of solving Equation 1 with different choices of $f$ is particularly important in the applications that we intend to consider, as it is interpreted as the initialization of the model. While various models employ iterative procedures for operator learning tasks implicitly, they lack a general theoretical perspective that justifies their approach. Several other models can be modified using iterative approaches to produce better performance with lower number of parameters. We will give experimental results in this regard to validate the practical benefit of our theoretical framework.

While the iterations considered so far have a fixed procedure which is identical per iteration, more general iterative procedures where the step changes between iterations are also diffused, and this can be done also adaptively.

### 3.4 APPLICATIONS

**Significance and Implications.** Our results underscore the existence of a solution for Equation 1 under certain conditions. Moreover, when the operator $T$ is bounded, our iterative method showcases uniform convergence. It follows that ensuring that the operators approximated by deep neural network architectures are contractive, we can introduce an iterative procedure that will allow us to converge to the fixed point as in Equation 2.

**Iterative Methods in Modern Deep Learning.** In contemporary deep learning architectures, especially those like Transformers, Stable Diffusion, AlphaFold, and Neural Integral Equations, the importance of operator learning is growing. However, these models, despite employing iterative techniques, often lack the foundational theoretical perspective that our framework provides. We will subsequently present experimental results that vouch for the efficacy and practical advantages of our theoretical insights.

**Beyond Basic Iterations.** While we have discussed iterations with fixed procedures, it is imperative to highlight that more general iterative procedures exist, and they can adapt dynamically. Further, there exist methods to enhance the rate of convergence of iterative procedures, and our framework is compatible with them.

## 4 NEURAL NETWORK ARCHITECTURES AS ITERATIVE OPERATOR EQUATIONS

In this section, we explore how various popular neural network architectures align with the framework of iterative operator learning. By emphasizing this operator-centric view, we unveil new avenues for model enhancements. Notably, shifting from implicit to explicit iterations can enhance model efficacy, i.e. through shared parameters across layers. A detailed discussion of the various methodologies given in this section is reported in Appendix B. In the appendix, we investigate architectures such as neural integral equations, transformers, AlphaFold for protein structure prediction, diffusion models, graph neural networks, autoregressive models, and variational autoencoders. We highlight the iterative numerical techniques underpinning these models, emphasizing potential advancements via methods like warm restarts and adaptive solvers. Empirical results substantiate the benefits of this unified perspective in terms of accuracy and convergence speed.

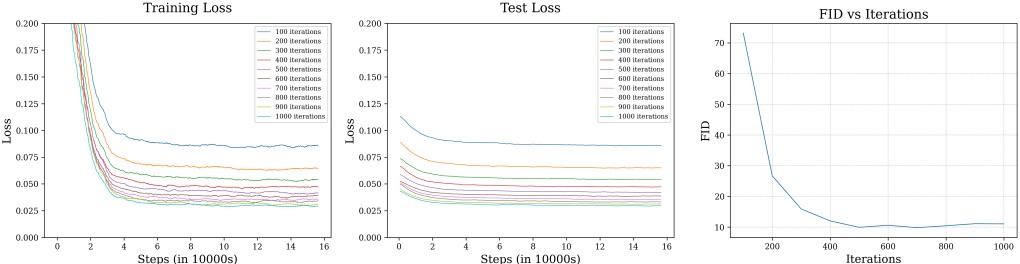

Figure 2: **Left and Middle**: Losses always decrease with more iterations in DDPMs. Training is stable and overfitting never occurs. EMA smoothing with $\alpha = 0.1$ is used for the loss curves to make the differences clearer. **Right**: DDPMs show FID and loss improves with an increased number of iterations on CIFAR-10. The number of iterations represent the denoising steps during training and inference. All diffusion models, UNets of identical architecture, are trained on CIFAR-10's training dataset with 64-image batches.

**Diffusion models.** Diffusion models, especially denoising diffusion probabilistic models (DDPMs), capture a noise process and its reverse (denoising) trajectory. While score matching with Langevin dynamics models (SMLDs) is relevant, our focus is primarily on DDPMs for their simpler setup. These models transition from complex pixel-space distributions to more tractable Gaussian distributions. Notably, increasing iterations can enhance the generative quality of DDPMs, a connection we wish to deepen. This procedure can be seen as instantiating an iteration procedure, where iterations are modified as in the methods found in Wolfe (1978). This operator setting and iterative interpretation is described in detail in Appendix B.1.

To empirically explore convergence with iterations in diffusion models, we train 10 different DDPMs with 100-1000 iterations and analyze their training dynamics and perceptual quality. Figure 2 reveals that increasing timesteps improves FID scores of generated images. Additionally, Figure 2 demonstrates a consistent decrease in both training and test loss with more time steps, attributed to the diminished area under the expected KL divergence curve over time (Figure 8). Notably, FID scores decline beyond the point of test loss convergence, stabilizing after approximately 150,000 steps (Figure 8). This behavior indicates robust convergence with increasing iterations.

**AlphaFold.** AlphaFold, a revolutionary protein structure prediction model, takes amino acid sequences and predicts their three-dimensional structure. While the model's intricacies can be found in Jumper et al. (2021), our primary interest lies in mapping AlphaFold within the operator learning context. Considering an input amino acid sequence, it undergoes processing to yield a multiple sequence alignment (MSA) and a pairwise feature representation. These data are subsequently fed into Evoformers and Structure Modules, iteratively refining the protein's predicted structure. We can think of the output of the Evoformer model as pair of functions lying in some discretized Banach space, while the Structure Modules of AlphaFold can be thought of as being operators over a space of matrices. This is described in detail in Appendix B.2.

To empirically explore the convergence behavior of AlphaFold as a function of iterations, we applied AlphaFold-Multimer across a range of 0-20 recycles on each of the 29 monomers using ground truth targets from CASP15. Figure 3 presents the summarized results, which show that while on average the GDT scores and RMSD improve with AlphaFold-Multimer, not all individual targets consistently converge, as depicted in Figures 4 and 5. Given that AlphaFold lacks a convergence constraint in its training, its predictions can exhibit variability across iterations.

**Graph Neural Networks.** Graph neural networks (GNNs) excel in managing graph-structured data by harnessing a differentiable message-passing mechanism. This enables the network to assimilate information from neighboring nodes to enhance their representations. We can think of the feature spaces as being Banach spaces of functions, which are discretized according to some grid. The GNN architecture can be thought of as being an operator acting on the direct sum of the Banach spaces, where the underlying geometric structure of the graph determines how the operator combines information through the topological information of the graph. A detailed description is given

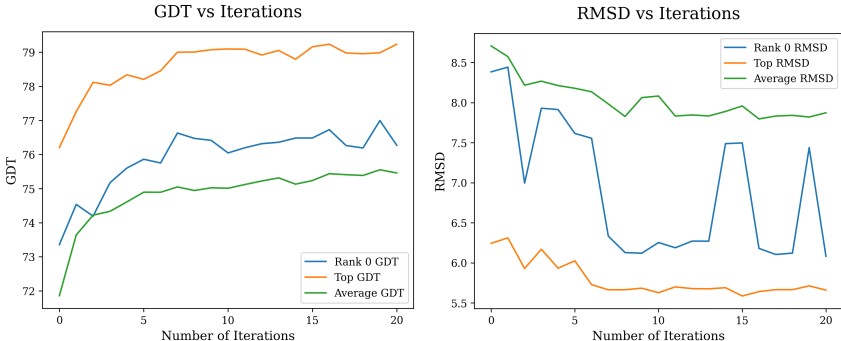

Figure 3: On average, additional iterations enhance AlphaFold-Multimer's performance, though the model doesn not invariably converge with more iterations. Target-specific trends can be seen in Figures 4 and 5.

in Appendix A.3, where theoretical guarantees for the convergence of the iterations are given, and Appendix B.3.

**Neural Integral Equations.** Neural Integral Equations (NIEs), and their variant Attentional Neural Integral Equations (ANIEs), draw inspiration from integral equations. Here, an integral operator, determined by a neural network, plays a pivotal role.

Denoting the integrand of the integral operator as $G_\theta$ within an NIE, the equation becomes:

$$\mathbf{y} = f(\mathbf{y}, \mathbf{x}, t) + \int_{\Omega \times [0,1]} G_\theta(\mathbf{y}, \mathbf{x}, \mathbf{z}, t, s) d\mathbf{z} ds$$

To solve such integral equations, one very often uses iterative methods, as done in Zappala et al. (2022) and the training of the NIE model consists in finding the parameters $\theta$ such that the solutions of the corresponding integral equations model the given data. A more detailed discussion of this model is given in Appendix B.4.

## 5 Experiments

In this section, we showcase experiments highlighting the advantages of explicit iterations. We introduce a new GNN architecture based on Picard iteration and enhance vision transformers with Picard iteration.

### 5.1 PIGN: Picard Iterative Graph Neural Network

To showcase the benefits of explicit iterations in GNNs, we developed **P**icard **I**teration **G**raph neural **N**etwork (PIGN), a GNN that applies Picard iterations for message passing. We evaluate PIGN against state-of-the-art GNN methods and another iterative approach called IterGNN (Tang et al., 2020) on node classification tasks.

GNNs can suffer from over-smoothing and over-squashing, limiting their ability to capture long-range dependencies in graphs (Nguyen et al., 2023). We assess model performance on noisy citation graphs (Cora and CiteSeer) with added drop-in noise. Drop-in noise involves increasing a percentage $p$ of the bag-of-words feature values, hindering classification. We also evaluate on a long-range benchmark (LRGB) for graph learning (Dwivedi et al., 2022).

Table 5 shows PIGN substantially improves accuracy over baselines on noisy citation graphs. The explicit iterative process enhances robustness. Table 1 illustrates PIGN outperforms prior iterative and non-iterative GNNs on the long-range LRGB benchmark, using various standard architectures. Applying Picard iterations enables modeling longer-range interactions.

| | **GCN** (Kipf & Welling, 2017) | **GAT** (Veličković et al., 2018) | **GraphSAGE** (Hamilton et al., 2017b) |
|---|---|---|---|
| w/o iterations | $0.1510 \pm 0.0029$ | $0.1204 \pm 0.0127$ | $0.3015 \pm 0.0032$ |
| IterGNN | $0.1736 \pm 0.0311$ | $0.1099 \pm 0.0459$ | $0.1816 \pm 0.0014$ |
| PIGN (Ours) | $\mathbf{0.1831 \pm 0.0038}$ | $\mathbf{0.1706 \pm 0.0046}$ | $\mathbf{0.3560 \pm 0.0037}$ |

Table 1: F1 scores of different models on the standard test split of the LRGB PascalVOC-SP dataset. Rows refer to model frameworks and columns are GNN backbone layers. A budget of 500k trainable parameters is set for each model. Each model is run on the same set of 5 random seeds. The mean and standard deviation are reported. For IterGNN with GAT backbone, two of the runs keep producing exploding loss so the reported statistics only include three runs.

The PIGN experiments demonstrate the benefits of explicit iterative operator learning. Targeting weaknesses of standard GNN training, PIGN effectively handles noise and long-range dependencies. A theoretical study of convergence guarantees is given in Appendix A.3.

---

**Algorithm 1** Picard Iteration Graph Neural Network (PIGN)

---

**Require:**
 $f$: Backbone GNN block
 Smoothing factor: $\alpha \in [0, 1]$
 Max number of iterations: $n$
 Input graph: $G = (V, E)$ with node features $X$
 Convergence threshold: $\epsilon$
1: $x_0 \leftarrow X$                    $\triangleright$ Initilization
2: $k \leftarrow 0$
3: **while** $\|x_{k+1} - x_k\| > \epsilon$ & $k < n$ **do**
4:   $z_{k+1} \leftarrow f(G, x_k)$        $\triangleright$ One iteration of Message Passing
5:   $x_{k+1} \leftarrow \alpha x_k + (1 - \alpha) z_{k+1}$   $\triangleright$ Update the node embedding with smoothing
6:   $k \leftarrow k + 1$
7: **return** $x_n$

---

## 5.2 ENHANCING TRANSFORMERS WITH PICARD ITERATION

We hypothesize that many neural network frameworks can benefit from Picard iterations. Here, we empirically explore adding iterations to Vision Transformers. Specifically, we demonstrate the benefits of explicit Picard teration in transformer models on the task of solving the Navier-Stokes partial differential equation (PDE) as well as self-supervised masked prediction of images. We evaluate various Vision Transformer (ViT) architectures (Dosovitskiy et al., 2020) along with Attentional Neural Integral Equations (ANIE) (Zappala et al., 2022).

For each model, we perform training and evaluation with different numbers of Picard iterations as described in Section 3.2. We empirically observe improved performance with more iterations for all models, since additional steps help better approximate solutions to the operator equations.

Table 2 shows lower mean squared error on the PDE task for Vision Transformers when using up to three iterations compared to the standard single-pass models. Table 3 shows a similar trend for self-supervised masked prediction of images. Finally, Table 4 illustrates that higher numbers of iterations in ANIE solvers consistently reduces error. We observe in our experiments across several transformer-based models and datasets that, generally, more iterations improve performance.

Overall, these experiments highlight the benefits of explicit iterative operator learning. For transformer-based architectures, repeating model application enhances convergence to desired solutions. Our unified perspective enables analyzing and improving networks from across domains. A theoretical study of the convergence guarantees of the iterations is given in Appendix A.2.

| Model | $N_{\text{iter}} = 1$ | $N_{\text{iter}} = 2$ | $N_{\text{iter}} = 3$ |
|---|---|---|---|
| ViT | $0.2472 \pm 0.0026$ | $0.2121 \pm 0.0063$ | $\mathbf{0.0691 \pm 0.0024}$ |
| ViTsmall | $0.2471 \pm 0.0025$ | $0.1672 \pm 0.0087$ | $\mathbf{0.0648 \pm 0.0022}$ |
| ViTparallel | $0.2474 \pm 0.0027$ | $0.2172 \pm 0.0066$ | $\mathbf{0.2079 \pm 0.0194}$ |
| ViT3D | $0.2512 \pm 0.0082$ | $\mathbf{0.2237 \pm 0.0196}$ | $0.2529 \pm 00.0079$ |

Table 2: ViT models used to solve a PDE (Navier-Stokes). The mean squared error is reported for each model as the number of iterations varies. A single iteration indicates the baseline ViT model. Higher iterations perform better than the regular ViT ($N_{\text{iter}} = 1$).

| Model | $N_{\text{iter}} = 1$ | $N_{\text{iter}} = 2$ | $N_{\text{iter}} = 3$ |
|---|---|---|---|
| ViT (MSE) | $0.0126 \pm 0.0006$ | $\mathbf{0.0121 \pm 0.0006}$ | $0.0122 \pm 0.0006$ |
| ViT (FID) | $20.0433$ | $20.0212$ | $\mathbf{19.2956}$ |

Table 3: ViT models trained with a pixel dropout reconstruction objective on CIFAR-10. The ViT architecture contains 12 encoder layers, 4 decoder layers, 3 attention heads in both the encoder and decoder. The embedding dimension and patch size are 192 and 2. The employed loss is MSE$((1 - \lambda)T(x_i) + \lambda x_i, y)$, computed on the final iteration $N_{\text{iter}} = i$. Images are altered by blacking out 75% of pixels. During inference, iterative solutions are defined as $x_{i+1} = (1 - \lambda)T(x_i) + \lambda x_i$, for $i \in \{0, 1, \ldots N\}$. Here, $N = 2$ and $\lambda = 1/2$.

| Model size | $N_{\text{iter}} = 1$ | $N_{\text{iter}} = 2$ | $N_{\text{iter}} = 4$ | $N_{\text{iter}} = 6$ | $N_{\text{iter}} = 8$ |
|---|---|---|---|---|---|
| $1H|1B$ | $0.0564 \pm 0.0070$ | $0.0474 \pm 0.0065$ | $0.0448 \pm 0.0062$ | $0.0446 \pm 0.0065$ | $\mathbf{0.0442 \pm 0.0065}$ |
| $4H|1B$ | $0.0610 \pm 0.0078$ | $0.0516 \pm 0.0083$ | $0.0512 \pm 0.0070$ | $0.0480 \pm 0.0066$ | $\mathbf{0.0478 \pm 0.0066}$ |
| $2H|2B$ | $0.0476 \pm 0.0065$ | $0.0465 \pm 0.0067$ | $0.0458 \pm 0.0067$ | $0.0451 \pm 0.0064$ | $\mathbf{0.0439 \pm 0.0062}$ |
| $4H|4B$ | $0.0458 \pm 0.0062$ | $0.0461 \pm 0.0065$ | $0.0453 \pm 0.0063$ | $0.0453 \pm 0.0061$ | $\mathbf{0.0445 \pm 0.0059}$ |

Table 4: Performance of ANIE on a PDE (Navier-Stokes) as the number of iterations of the integral equation solver varies, and for different sizes of architecture. Here $H$ indicates the number of heads and $B$ indicates the number of blocks (layers). A single iteration means that the integral operator is applied once. As the number of iterations of the solver increases, the performance of the model in terms of mean squared error improves.

## 6 DISCUSSION

We introduced an iterative operator learning framework in neural networks, drawing connections between deep learning and numerical analysis. Viewing networks as operators and employing techniques like Picard iteration, we established convergence guarantees. Our empirical results, exemplified by PIGN—an iterative GNN, as well as an iterative vision transformer, underscore the benefits of explicit iterations in modern architectures.

For future work, a deeper analysis is crucial to pinpoint the conditions necessary for convergence and stability within our iterative paradigm. There remain unanswered theoretical elements about dynamics and generalization. Designing network architectures inherently tailored for iterative processes might allow for a more effective utilization of insights from numerical analysis. We are also intrigued by the potential of adaptive solvers that modify the operator during training, as these could offer notable advantages in both efficiency and flexibility.

In summation, this work shines a light on the synergies between deep learning and numerical analysis, suggesting that the operator-centric viewpoint could foster future innovations in the theory and practical applications of deep neural networks.

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

## A   THEORETICAL GUARANTEES ON CONVERGENCE

In this section we consider the problem of convergence of the iterations for certain classes of models considered in the article.

### A.1   GENERALITIES ON OPERATOR DIFFERENTIATION

We recall some general facts about Frechet and Gateaux differentiation of operators between Banach spaces which will be used later in the article to provide criteria for the convergence of the iterations.

**Definition 1.** *Let $X, Y$ be a Banach spaces, let $T : X \longrightarrow Y$ be a continuous mapping, and let $u \in X$ be an arbitrary element in the domain of $T$. We say that $T$ is Frechet differentiable at $u$ if it holds that*

$$T(u + h) = T(u) + Ah + o(||h||) \qquad (6)$$

*for some linear operator $A : X \longrightarrow Y$, for $h \to 0$. In such a situation we say that $A$ is the Frechet derivative of $T$ at $u$, and denote the operator $A$ by the symbol $T'(u)$.*

*We say that $T$ is Gateaux differentiable at $u$ if it holds that*

$$\lim_{t \to 0} \frac{T(u + th) - T(u)}{t} = Ah \qquad (7)$$

*for some linear operator $A : X \longrightarrow Y$, and for all $h \in X$. In such a situation we say that $A$ is the Gateaux derivative of $T$ at $u$, and denote the operator $A$ by the symbol $T'(u)$.*

**Remark 1.** Observe that the derivative is a mapping from the domain of $T$ to the space of linear maps $X \longrightarrow Y$, which we denote (following common notational convention) by $L(X, Y)$. The latter is a Banach space whenever $Y$ is a Banach space, where the norm is the usual norm of linear operators.

Of course, the notion of Frechet derivative is stronger than that of Gateaux derivative. In other words, if $T$ is Frechet differentiable, then it is also Gateax differentiable. The converse is not always true if the spaces $X$ and $Y$ are infinite dimensional.

Differentiability of operators is a fundamental notion in functional analysis. We recall here an important property, namely the Mean Value Theorem.

**Proposition 1.** *(Mean Value Theorem for operators) Let $X, Y$ be Banach spaces, and assume that $T : U \subset X \longrightarrow Y$ is an operator where $U \subset X$ is an open set of $X$. Suppose that $T$ is (either Frechet or Gateaux) differentiable on $U$ and that $T' : U \longrightarrow L(X, Y)$ is a continuous mapping. Let $u, v \in U$ and assume that the segment between $u$ and $v$ lies in $U$. Then we have*

$$||T(u) - T(v)||_Y \leq \sup_{\theta \in [0,1]} ||T'((1-\theta)u + \theta w)|| \, ||u - w||_X. \tag{8}$$

As a consequence, we obtain the following useful criterion for an operator to be Lipschitz.

**Lemma 1.** *Let $T : U \subset X \longrightarrow Y$ be an operator betweem Banach spaces that is (either Frechet or Gateaux) differentiable with continuous derivative over the open and convex $U$. If $||T'(x)|| \leq M$ for all $x \in U$ and some $M > 0$, then $T$ is Lipschitz continuous on $U$ with Lipschitz constant at most $M$.*

*Proof.* Let $u, w$ be arbitrary elements of $U$. Since $U$ is convex, the line segment between $u$ and $w$ is inside $U$:

$$(1 - \theta)u + \theta v \in U$$

for all $\theta \in [0, 1]$, which by assumption implies that $||T'((1 - \theta)u + \theta v)|| \leq M$. From the Mean Value Theorem for operators (Proposition 1) we have that

$$||T(u) - T(v)|| \leq \sup_{\theta \in [0,1]} ||T'((1-\theta)u + \theta v)|| \, ||u - v||_X \leq M \cdot ||u - v||_X, \tag{9}$$

from which it follows that $T$ is Lipschitz continuous on $U$ with Lipschitz constant at most $M$. $\square$

### A.2 ITERATIONS AND TRANSFORMER MODELS

While iterative approaches as those in ANIE have been explicitly used to solve a corresponding fixed-point type of problem for a transformer operator that mimics integration, one can imagine to generalize such a procedure to any transformer architecture, and iterate through the same transformer several times during training and evaluation.

This formulation of transformer models, as operators whose corresponding Equation 1 we want to solve, is inspired by integral equation methods, and will be shown to give better performance with fixed number of parameters in the experiments. In fact, one similar approach has been followed for some transformer architectures in the diffusion models, as we discuss in Subsection 4.

We consider a transformer consisting of a single self-attention layer. A generalization to multiple layers is obtained straightforwardly from the considerations given in this subsection. We want to show that Theorem 1 is applicable when considering iterations of a transformer architecture.

Here we set $T : X \longrightarrow X$ to be a single layer transformer architecture consisting of a self-attention mechanism with matrices $W^Q, W^K, W^V$, which are known in the literature as queries, keys and values, respectively.

First, we will show that transformers are Frechet and Gateaux differentiable. Then, making use of Lemma 1 we will find a criterion to force the iteration method applied to a transformer to converge to a solution of an equation of the same type as in Equation 1. Here we consider the space $X$ of continuous (and therefore bounded) functions $[0, 1] \longrightarrow \mathbf{R}$. We assume that the space is appropriately discretized through a grid of points $\{t_i\}_{i=0}^{N-1} \subset [0, 1]$ with $t_0 = 0$ and $t_{N-1} = 1$. The same

reasoning can be applied to discretizations of higher dimensional domains, but we will explicitly consider only this simpler situation. Here a function $f$ is given by a sequence of points $f(t_i)$ which coincides with the value of $f$ on $t_i$ for all $i = 0, \ldots, N-1$.

**Lemma 2.** *Let $T$ denote a single layer transformer architecture where $W^Q, W^K, W^V$ denote its query, key and value matrices. Then $T$ is Frechet (hence Gateaux) differentiable.*

*Proof.* Let $y$ be a discretized function in $X$, i.e. $y \approx \{y(t_i)\}$. Then we have

$$T(y) = (yW^Q)(yW^K)^T(yW^V) = W_y K_y^T V_y.$$

We observe that this is cubic in the input $y$. Let us consider $h \in X$ which is a discretized function as well. Then we can write

$$
\begin{aligned}
T(y+h) &= ((y+h)W^Q)((y+h)W^K)^T((y+h)W^V) && (10)\\
&= (yW^Q)(yW^K)^T(yW^V) + (hW^Q)(yW^K)^T(yW^V) && (11)\\
&\quad + (yW^Q)(hW^K)^T(yW^V) + (yW^Q)(yW^K)^T(hW^V) && (12)\\
&\quad + R(h) && (13)\\
&= T(y) + T_1'(y)(h) + T_2'(y)(h) + T_3'(y)(h) + R(h), && (14)
\end{aligned}
$$

where $R(h)$ contains terms that are quadratic and cubic in $h$, and we have set $T_i'(y)(h)$ to be the components that are linear in $h$ and where $h$ appears in the position $i$ with respect to the product $QK^TV$. Therefore, we have written $T(y+h)$ as $T(y)$ plus terms that are linear in $h$, and a remainder that is at least quadratic in $h$, and that it is therefore an $o(||h||)$ when $h \longrightarrow 0$. It follows that $T$ is Frechet at $y$, and since $y$ was arbitrarily chosen, $T$ is Frechet in the whole domain. $\qquad\square$

**Theorem 3.** *Let $T$ denote a single layer transformer such that its Frechet derivative has norm bounded by some $M$ such that $\lambda M < 1$ for all $y \in U$, where $U$ is defined to be the unit ball around $0$ in $X$. Then, Equation 1 admits a solution for any initialization $f$, and this is the limit of the iterations $y_k := f + T(y_{k-1})$ for all $k > 1$ with $y_0 := f$.*

*Proof.* We observe first that by applying Lemma 2 it follows that $T$ admits Frechet derivative everywhere on its domain, so that $T'$ exists, and the assumption in the statement of this theorem is meaningful. Since $M < 1/\lambda$, it follows from Lemma 1 that $T$ is Lipschitz over $U$ with Lipschitz constant $L$ such that $\lambda L < 1$. It follows that the assumptions in Theorem 1 and its corollaries hold and the result now follows. $\qquad\square$

In practice, in order to enforce the convergence of the iterative method to a solution of Equation 1 we can add a term in the loss where the Frechet derivative as computed in Lemma 2 appears explicitly.

The algorithm for the iterations takes in the case of transformers a similar form as in the case of Algorithm 1. This is given in Algorithm 2.

---

**Algorithm 2** Iterative procedure for solving Equation 1 with a transformer based model $T$

---

**Require:**
    $T$: Transformer architecture
    $f$: Free function in Equation (1)
    Smoothing factor: $\alpha \in [0, 1]$
    Max number of iterations: $n$
1:  $x_0 \leftarrow f$
2:  $k \leftarrow 0$
3:  **while** $||x_{k+1} - x_k|| > \epsilon$ & $k < n$ **do**
4:     $z_{k+1} \leftarrow T(x_k) + f$                                              ▷ Iteration
5:     $x_{k+1} \leftarrow \alpha x_k + (1-\alpha)z_{k+1}$   ▷ Update the new solution using the previous two iterations and smoothing
6:     $k \leftarrow k+1$
7:  **return** $x_n$

---

### A.3 GRAPH NEURAL NETWORKS

We now consider the case of GNNs, and reformulate them as an operator learning problem in some space of functions. We analyze one well known type of GNN architecture and show that under certain constraints, the convergence of the iterations is guaranteed.

First, recall that a GNN hads a geometric support, the graph $\Gamma = \{V_\Gamma, E_\Gamma\}$, where $V_\Gamma$ is the set of vertices and $E_\Gamma$ is the set of edges, along with spaces of features $X_v$ for each $v \in V_\Gamma$. The neural network $T$ acts on the spaces of features based on the geometric information given by $E_\Gamma$, which determine the neighborhoods of the vertices. Here we consider a slightly more general situation where we have a copy of the Hilbert space $X_v = L_2([-1, 1])$ associated to each vertex $v$, and we define $T$ to be an operator on the direct sum of spaces $X = \bigoplus_{v \in E_\Gamma} X_v$. The operator $T$ uses the geometric information of $E_\Gamma$ to act on $X$. The case of neural networks, in practice, can be thought of as a case where the function spaces $L_2([-1, 1])$ are discretized, giving rise to the feature spaces. The discussion that follows can be recovered adapted to the discretized case as well. More generally, one can think of each $X_v$ as being a Banach space of functions, but we will not discuss this more general case in detail here.

At each vertex, GNNs apply an aggregate function that depends on the architecture used. A common choice is to use some pooling function such as applying a matrix $W$ to each feature vector on a neighborhood of the vertex $v$, use a RELU nonlinearity, and take the maximum (component-wise) over the outputs. See Hamilton et al. (2017a). This operation is followed by some combination function, which in the variant Hamilton et al. (2017a) that we are considering, usually consists of concatenation (followed by a linear map). We translate the aggregate function described above into operators over the direct sum Hilbert space $X$. Since our discussion can be directly extended to operators including the combine function as well, we will consider our operator $T$ as consisting only of the aggregation functions. We indicate the aggregation operator at node $v \in V_\Gamma$ by $A_v$. This is obtained as a composition of a linear operator $W : L_2([-1, 1]) \longrightarrow L_2([-1, 1])$ (a matrix in the discretized form), which is applied over all the function in $X_{v_i}$ for all $v_i \in N(v)$, the neighborhood of $v$. This is followed by the mapping $R$, which is performs RELU nonlinearity over the entries of the function output of $W$ component-wise. Finally, we take the maximum over $i$ of component-wise. This procedure is a direct generalization to our setting of the common pooling aggregation function, which is recovered when we discretize the space using a grid in the domain $[-1, 1]$. The linear map $W$, here, is assumed to be shared among operators $A_v$, but our discussion can be straightforwardly adapted to the case of non-shared linear maps $W$.

Recall that an element of the direct sum of Hilbert spaces $X = \bigoplus_i X_i$ takes the form $f = f_1 + \cdots + f_k$, where $k = |V_\Gamma|$. We write the operator $A_v$ as

$$A_v(f)(t) = \max_i \{R(W(f_i))(t) \mid v_i \in N(v)\}. \tag{15}$$

Note that the dependence of $A_v$ on the edges of the geometric support $\Gamma$ gets manifested through the neighborhood of $v$, $N(v)$. The GNN operator $T$, then, is given by

$$T(f) = \sum_i \pi_i T(f) = \sum_i A_{v_i}(f), \tag{16}$$

where $\pi_i : X \longrightarrow X_{v_i}$ is the canonical projection and each summand is defined through Equation 15.

Before finding explicit guarantees on the convergence of the iterations for GNNs, we have the following useful result.

**Lemma 3.** *Let $T$ denote the GNN operator described above, let $\Gamma$ be its geometric support, and let $X$ be the total Hilbert space. Assume that $W$ is a Lipschitz operator with constant $L$. Then, there exists a polynomial of degree 1 with non-negative coefficients, $P(z_1, \ldots, z_k) \in \mathbb{Z}[z_1, \ldots, z_k]$, such that the following inequality holds*

$$||T(f) - T(g)|| \leq L \cdot P(||f_1 - g_1||, \ldots, ||f_k - g_k||), \tag{17}$$

*where $f = \sum_i f_i$ and $g = \sum_i g_i$, with $f_i, g_i \in X_{v_i}$ for all $i = 1, \ldots, k$. Moreover, $P$ depends only on $\Gamma$.*

*Proof.* By definition of norm in the direct sum space, and considering the fact that $T$ decomposes into a direct sum as in Equation 16, we have

$$||T(f) - T(g)|| \quad = \quad ||\bigoplus_v A_v(f) - \bigoplus_v A_v(g)|| \tag{18}$$

$$= \quad ||\bigoplus_v [A_v(f) - A_v(g)]|| \tag{19}$$

$$= \quad \bigoplus_v ||A_v(f) - A_v(g)||. \tag{20}$$

We consider therefore the term $||A_v(f) - A_v(g)||$ for an arbitrary $v \in V_\Gamma$. It holds

$$||A_v(f) - A_v(g)|| \quad = \quad ||\max_i(RW(f_i)) - \max_i(RW(g_i))|| \tag{21}$$

$$\leq \quad ||\max_i(RW(f_i) - RW(g_i))|| \tag{22}$$

$$\leq \quad \sum_i ||RW(f_i) - RW(g_i)|| \tag{23}$$

$$\leq \quad \sum_i L||f_i - g_i||, \tag{24}$$

where we have used the fact that if $W$ is Lipschitz with constant $L$, then $RW$ is also Lipschitz with constant at most $L$ (since RELU only zeroes the negative part). Therefore, we have

$$||T(f) - T(g)|| \leq \sum_v \sum_i L||f_i - g_i||. \tag{25}$$

Some of the $||f_i - g_i||$ appear in multiple $v$'s, depending on the nieghborhoods determined by $E_\Gamma$. However, this is a degree 1 polynomial in the $||f_i - g_i||$'s, and since it is determined uniquely by the graph $\Gamma$, we have completed the proof. $\square$

Now we can show that when the maps $W$ are contractive, the iterations are guaranteed to converge.

**Theorem 4.** *Let $T$ be a GNN operator acting on $X$ as above. Suppose $W$ is Lipschitz and let $\alpha := \max_i \alpha_i$, where $\alpha_i$ are the coefficient of the polynomial $P$ in Lemma 3, such that $L\alpha < 1$. Then, the fixed point problem*

$$T(y) + f = y$$

*has a unique solution for any choice of $f \in X$, and the iterations $y_n$ converge to such solution, where $y_0 = f$ and $y_{n+1} = T(y_n) + f$.*

*Proof.* Using Lemma 3 and the hypothesis, we can reduce this result to an application of Theorem 1. In fact, let $P(z_1, \ldots, z_k) = \alpha_1 z_1 + \cdots, \alpha_k z_k$. Then, from Lemma 3 we have

$$||T(f) - T(g)|| \quad \leq \quad L(\alpha_1 ||f_1 - g_1|| + \cdots + \alpha_k ||f_k - g_k||) \tag{26}$$

$$\leq \quad L\alpha(||f_1 - g_1|| + \cdots + ||f_k - g_k||) \tag{27}$$

$$= \quad L\alpha ||f - g||. \tag{28}$$

Since $L\alpha < 1$ we have that $T$ is contractive in the direct sum Hilbert space $X$, and we can apply the previous machinery to enforce convergence and uniqueness. This result is independent of $f$ as in Corollay 2. $\square$

While we have considered here the specific architecture of SAGE Hamilton et al. (2017a), a similar approach can be pursued in general for aggregation functions, at least with well behaved pooling functions.

# B  NEURAL NETWORK ARCHITECTURES AS ITERATIVE OPERATOR EQUATIONS IN DETAIL

This section underscores how some prominent deep learning models fit within our iterative operator learning framework. While many architectures employ implicit iterations, transitioning to an explicit approach has proven to yield improvements in model efficacy and data efficiency.

We delve into a range of architectures including neural integral equations, transformers, AlphaFold for protein structure prediction, diffusion models, graph neural networks, autoregressive models, and variational autoencoders. For each, we elucidate their alignment with the operator perspective and the compatibility with our framework.

Recognizing the underlying iterative numerical procedures in these models grants opportunities for enhancements using strategies such as warm restarts and adaptive solvers. Empirical results on various datasets underscore improvements in accuracy and convergence speed, underscoring the merits of this unified approach. In summary, the iterative operator perspective merges theoretical robustness with tangible benefits in modern deep learning.

## B.1 DIFFUSION MODELS

Diffusion models Sohl-Dickstein et al. (2015) are a class of generative models that take a fixed noise process and learn the reverse (denoising) trajectory. Although our discussion applies to score matching with Langevin dynamics models (SMLDs) Song & Ermon (2019), we focus on denoising diffusion probabilistic models (DDPMs) Ho et al. (2020) due to their simpler setup. Typically, DDPMs are applied to images to learn a mapping from the intractable pixel-space distribution to the standard Gaussian distribution that can easily be sampled. It has been observed (and briefly shown in Kingma et al. (2021)) that more iterations improve the generative quality of DDPMs, and our aim is to thoroughly explore this connection.

Let us review the setup for DDPMs from Ho et al. (2020). For a fixed number of time steps $t = 0, 1, 2, \ldots, T$, diffusion schedule $\beta_1, \beta_2, \ldots, \beta_T$, and unknown pixel-space distribution $q(\mathbf{x}_0)$, the forward process is a Markov process with a Gaussian transition kernel

$$q(\mathbf{x}_t|\mathbf{x}_{t-1}) \sim \mathcal{N}(\sqrt{1 - \beta_t}\mathbf{x}_{t-1}, \beta_t\mathbf{I}).$$

Let $\alpha_t := 1 - \beta_t$ and $\bar{\alpha}_t := \prod_{i=1}^{t} \alpha_i$. Given $\mathbf{x}_0$, we can deduce from the Markov property and the form of $q(\mathbf{x}_t|\mathbf{x}_{t-1})$ a closed form for $q(\mathbf{x}_t|\mathbf{x}_0)$:

$$q(\mathbf{x}_t|\mathbf{x}_0) \sim \mathcal{N}(\sqrt{\bar{\alpha}_t}\mathbf{x}_0, (1 - \bar{\alpha}_t)\mathbf{I}).$$

A sample from this distribution can be reparametrized as

$$\mathbf{x}_t = \sqrt{\bar{\alpha}_t}\mathbf{x}_0 + \sqrt{1 - \bar{\alpha}_t}\boldsymbol{\epsilon}, \quad \boldsymbol{\epsilon} \sim \mathcal{N}(\mathbf{0}, \mathbf{I}).$$

Since $q(\mathbf{x}_0)$ is unknown, the reverse process $q(\mathbf{x}_{t-1}|\mathbf{x}_t)$ cannot be analytically derived from Bayes' rule so it must be approximated. The goal of a DDPM is to approximate the reverse process by learning a noise model $\boldsymbol{\epsilon}_\theta(\mathbf{x}_t, t)$ that predicts $\boldsymbol{\epsilon}$ from the noisy image $\mathbf{x}_t$ at time step $t$. During training, a timestep $t$ is chosen uniformly at random, and then the loss is $C(t)\|\boldsymbol{\epsilon} - \boldsymbol{\epsilon}_\theta(\mathbf{x}_t, t)\|^2$, where $C(t)$ is some weighting depending on $t$ or equal to 1. At inference, the denoising trajectory is computed via Langevin sampling using the model as predictor of partial denoising at each time step.

The connection to operator learning is as follows (our discussion follows Song et al. (2021)). In the continuous setting, we can let $\beta(t)$ be the continuous diffusion schedule corresponding to the discrete diffusion schedule $\{\beta_i\}$, and let $\mathbf{w}$ be a Wiener process. In , it is shown that the forward SDE is given by

$$d\mathbf{x} = -\frac{1}{2}\beta(t)\mathbf{x}dt + \sqrt{\beta(t)}d\mathbf{w},$$

and the reverse SDE is

$$d\mathbf{x} = -\beta(t)\left(\frac{1}{2} + \nabla_\mathbf{x}\log p_t(\mathbf{x})\right)dt + \sqrt{\beta(t)}d\bar{\mathbf{w}},$$

where $p_t(\mathbf{x})$ is the probability distribution of $\mathbf{x}$ at time $t$ and time flows backwards. SMLDs and DDPMs learn a discretized version of the reverse SDE by approximating the score $\nabla_\mathbf{x}\log p_t(\mathbf{x})$, which is easily seen to be the KL divergence terms in the loss function of DDPMs up to a constant.

**Diffusion training details.** We use Hugging Face's von Platen et al. (2022) UNet2D model with 5 downsampling blocks and 5 upsampling blocks where the 4th downsampling and 2nd upsampling blocks contain attention blocks. Each block contains 2 ResNet2D blocks, and the number of

outchannels is (128, 128, 256, 256, 512, 512) from the first input convolutional layer up to the middle of the UNet, and the number of outchannels is reversed during upsampling. We set $\beta_1 = .95$, $\beta_2 = .999$, $\epsilon = 1e - 8$ for the Adam optimizer with learning rate 1e-4, weight decay 1e-6, and 500 warmup steps. The loss function is mean-square-error loss on noise prediction for a single uniformly sampled timestep for each image. The batch size is set to 64 images, and each model is trained up to 200,000 steps.

## B.2 ALPHAFOLD

AlphaFold is a deep learning approach to predict the three-dimensional structure of proteins based on given amino acid sequences Jumper et al. (2021). For the sake of this paper, we briefly explain the AlphaFold model and then formulate it in the context of an operator learning problem.

Each input amino acid sequence is first processed to create a multiple sequence alignments (MSA) representation of dimension $s \times r \times c$ and a pairwise feature representation of dimension $r \times r \times c$. Both representations are passed to repeated layers of Evoformers to encode physical, biological and geometric information into the representations. The representations are then passed to a sequence of weight-sharing Structure Modules to predict and iteratively refine the protein structure. The entire process is recycled for $N$ times in order to further refine the prediction. See Jumper et al. (2021) for details.

In the context of operator learning, we think of the input to the model as a pair of functions $(f, g)$, where $f : [0, 1] \to \mathbb{R}^c$ and $g : [0, 1] \times [0, 1] \to \mathbb{R}^c$. We discretize the interval $[0, 1]$ to a grid of $r$ points to represent the input functions so that they can be processed by the model.

Let $\mathcal{X}$ and $\mathcal{Y}$ denote the function spaces where $f$ and $g$ lives respectively. The entire sequence of Evoformer can be described as an operator

$$E : \mathcal{X} \times \mathcal{Y} \to \mathcal{X} \times \mathcal{Y}$$

Since the Structure Modules share the same weights, we consider each one of them as an operator $S$ and the sequence of $k$ Structure Modules as composing $S$ for $k$ times. We have

$$S : \mathcal{X} \times M_3(\mathbb{R}) \times \mathbb{R}^3 \to \mathcal{X} \times M_3(\mathbb{R}) \times \mathbb{R}^3$$

The output space of AlphaFold is $M_3(\mathbb{R}) \times \mathbb{R}^3$, consisting of pairs of rotation matrices and translation vectors. Denote $y = (f, g) \in \mathcal{X} \times \mathcal{Y}$. One cycle of the AlphaFold model can be described as an operator

$$AF : \mathcal{X} \times \mathcal{Y} \to M_3(\mathbb{R}) \times \mathbb{R}^3$$
$$y \mapsto S^k T(y)$$

AlphaFold does not impose any conditions to guarantee convergence of its recycles, but it shows convergence behavior at inference in Figure 3. We look at the GDT scores and RMSD for up to 20 iterations across 29 different monomeric proteins and 44 targets from the CASP15 dataset cas (2022). While convergence behavior is seen on average, Figures 4 and 5 suggest that AlphaFold may benefit from an explicit convergence constraint.

**AlphaFold inference details.** We use the full database and multimer settings at inference. 5 random seeds are chosen for each of the 5 models. We do not fix the random seeds—each prediction uses a different set of random seeds. Our max template date is set to 2022-01-01 to avoid leakage with CASP15. The targets were selected from CASP15 and filtered on public availability of targets.

## B.3 GRAPH NEURAL NETWORKS

Graph neural networks (GNNs) specialize in processing graph-structured data by utilizing a differentiable message-passing mechanism. This mechanism assimilates information from neighboring nodes to refine node representations.

For a given graph

$$\mathcal{G} = (\mathcal{V}, \mathcal{E}),$$

with $\mathcal{V}$ as the node set and $\mathcal{E}$ as the edge set, the feature vector of a node $v \in \mathcal{V}$ is represented by $\mathbf{x}_v$. A frequently used aggregation strategy is:

$$\mathbf{x}'_v = f(\mathbf{x}_v, \mathrm{AGG}_{\mathcal{N}(v)}(\{\mathbf{x}_u \mid u \in \mathcal{N}(v)\})) \tag{29}$$

Here, $\mathcal{N}(v)$ specifies the neighbors of node $v$, AGG is an aggregation function (e.g., sum or mean) applied over the neighboring nodes' features, and $f$ acts as a neural network-based update function.

Interpreting the aggregation within the framework of operator equations, let us denote $\mathcal{X}$ as $\mathbb{R}^{|\mathcal{V}| \times d}$, representing the space of node feature matrices. The aggregation and subsequent node feature updates can be described by an operator $T : \mathcal{X} \longrightarrow \mathcal{X}$, transforming the existing features into their refined versions. The equilibrium or fixed points $\mathbf{X}^*$ are achieved when:

$$\mathbf{X}^* = T(\mathbf{X}^*) \tag{30}$$

Training GNNs thus involves setting initial features as $\mathbf{X}_0$ and allowing the operator $T$ to iteratively refine them until convergence. This operator-centric viewpoint naturally extends to multi-layered architectures, reminiscent of the GraphSAGE approach, integrating seamlessly within the proposed framework. A more theoretical framework for this setup is given in Appendix A.3.

### B.4 NEURAL INTEGRAL EQUATIONS

Neural Integral Equations (NIEs), and their variation Attentional Neural Integral Equations (ANIEs), are a class of deep learning models based on integral equations (IEs) Zappala et al. (2022). Recall that an IE is a functional equation of type

$$\mathbf{y} = f(\mathbf{y}, \mathbf{x}, t) + \int_{\Omega \times [0,1]} G(\mathbf{y}, \mathbf{x}, \mathbf{z}, t, s) d\mathbf{z} ds, \tag{31}$$

where the integrand function $G$ is in general nonlinear in $\mathbf{y}$. Several types of IEs exist, and this class contains very important examples but it is not exhaustive.

If $G$ is contractive with respect to $\mathbf{y}$, then we can adapt the proof of Theorem 1 to show that the equation admits a unique solution $\mathbf{y}$, and the iteration thereby described converges to a solution. In fact, IEs are a class of equations where iterative methods have found important applications due to their non-local nature. More specifically, we observe that when solving an IE, this cannot be solved in a sequential manner, since in order to evaluate $\mathbf{y}$ at any point $(\mathbf{x}, t)$ we need to integrate $\mathbf{y}$ over the full domain $\Omega \times [0, 1]$, which requires knowledge of a solution globally. This is in stark contrast with ODEs, where knowledge of $\mathbf{y}(t)$ at one point determines $\mathbf{y}(t)$ at a "close" next point.

NIEs are equations as 31, where the integral operator $\int_{\Omega \times [0,1]} G(\bullet, \mathbf{x}, \mathbf{z}, t, s) d\mathbf{z} ds$ is determined by a neural network. More specifically, $G := G_\theta$ is a neural network, where $\theta$ indicates the parameters:

$$\mathbf{y} = f(\mathbf{y}, \mathbf{x}, t) + \int_{\Omega \times [0,1]} G_\theta(\mathbf{y}, \mathbf{x}, \mathbf{z}, t, s) d\mathbf{z} ds, \tag{32}$$

Optimization of an NIE consists in finding parameters $\theta$ such that the solutions $\mathbf{y}$ of Equation 32, with respect to different choices of $f$ called initialization, fit a given dataset. See Zappala et al. (2022) for details.

ANIEs are a more general approach to IEs based on transformers (Zappala et al. (2022)). The corresponding equation takes the form

$$T(\mathbf{y}) + f(\mathbf{y}) = \mathbf{y}, \tag{33}$$

where $T$ indicates a transformer architecture. In Zappala et al. (2022), it is argued that a self-attention layer can be seen as an integration operator of the type given in Equation 32, so that these models are conceptually the same as NIEs, but their convenience relies in the implementation of transformers. As an operator, $T$ here is assumed to map a function space into itself, where a discretization procedure has been applied on $\Omega \times [0, 1]$ to obtain functions on grids. The iterative method described in Theorem 1 is used for ANIEs as well, under the constraint that during training the transformer architecture determines a contractive mapping on $\mathbf{y}$.

|  | **Cora** | **Cora-50%** | **CiteSeer** | **CiteSeer-50%** |
|---|---|---|---|---|
| GCN | $\mathbf{80.9 \pm 0.6}$ | $33.2 \pm 12.2$ | $\mathbf{71.3 \pm 0.6}$ | $18.7 \pm 6.3$ |
| GAT | $\mathbf{80.4 \pm 1.1}$ | $56.3 \pm 4.4$ | $\mathbf{69.9 \pm 1.0}$ | $26.4 \pm 5.4$ |
| IterGNN with GCN | $60.9 \pm 17.4$ | $36.9 \pm 13.3$ | $45.33 \pm 10.06$ | $25.7 \pm 7.4$ |
| IterGNN with GAT | $60.5 \pm 17.5$ | $38.84 \pm 15.24$ | $49.91 \pm 4.94$ | $30.20 \pm 7.20$ |
| PIGN with GCN (Ours) | $76.7 \pm 1.4$ | $\mathbf{43.4 \pm 6.8}$ | $68.4 \pm 1.5$ | $\mathbf{31.0 \pm 3.1}$ |
| PIGN with GAT (Ours) | $79.2 \pm 0.9$ | $\mathbf{57.3 \pm 4.8}$ | $66.1 \pm 1.7$ | $\mathbf{34.3 \pm 4.6}$ |

Table 5: Accuracy scores of the three model frameworks with two GNN backbone layers on the standard test split of the original and noisy Cora and CiteSeer datasets. Each model is run for 100 times and mean and standard deviation are reported. Our PIGN framework achieves comparable performance on orginal datasets and better performance on noisy datasets.

Neural Integro-Differential Equations (NIDEs) are deep learning models closely related to NIEs and ANIEs, and are based on Integro-Differential Equations (IDEs) Zappala et al. (2023). The iterations for IDEs are conceptually similar to those discussed for IEs, but they also contain a differential solver step due to the presence of the differential operator in IDEs. We refer the reader to Zappala et al. (2023) for a more detailed treatment of the solver procedure.

## B.5 AUTOREGRESSIVE MODELS

Autoregressive models, inherent in sequence-based tasks, predict elements of a sequence by leveraging the history of prior elements. Given a sequence $\mathbf{x} = (x_1, x_2, ..., x_n)$, the joint distribution $p(\mathbf{x})$ is expressed as:

$$p(\mathbf{x}) = \prod_{i=1}^{n} p(x_i | x_{<i}) \tag{34}$$

where $x_{<i}$ represents the history, i.e., $(x_1, ..., x_{i-1})$. Each term $p(x_i | x_{<i})$ uses a neural architecture to predict the distribution of $x_i$ based on its antecedents.

Widely-recognized autoregressive networks like PixelCNN, used for images, and Transformer-based decoders like GPT for text, employ this approach. Training involves assessing the difference between predicted $p(x_i | x_{<i})$ and the actual $x_i$ from the dataset.

Incorporating the iterative operator framework, consider $\mathcal{X}$ as a functional space appropriate for sequences. This allows the definition of an operator $T : \mathcal{X} \to \mathcal{X}$ that iteratively updates the sequence:

$$T(\mathbf{x}) = (x_1, f_2(x_1), ..., f_n(x_{<n})) \tag{35}$$

Here, $f_i$ is the neural component predicting $p(x_i | x_{<i})$. The model aims for fixed points $\mathbf{x}^*$ where $\mathbf{x}^* = T(\mathbf{x}^*)$, equating predicted and actual subsequences.

By repeatedly applying $T$ to an initial sequence and utilizing metrics like the Kullback-Leibler divergence to measure proximity to fixed points, we establish a systematic approach to maximize the likelihood, aligning it with our framework.

## C MORE NUMERICAL RESULTS AND FIGURES

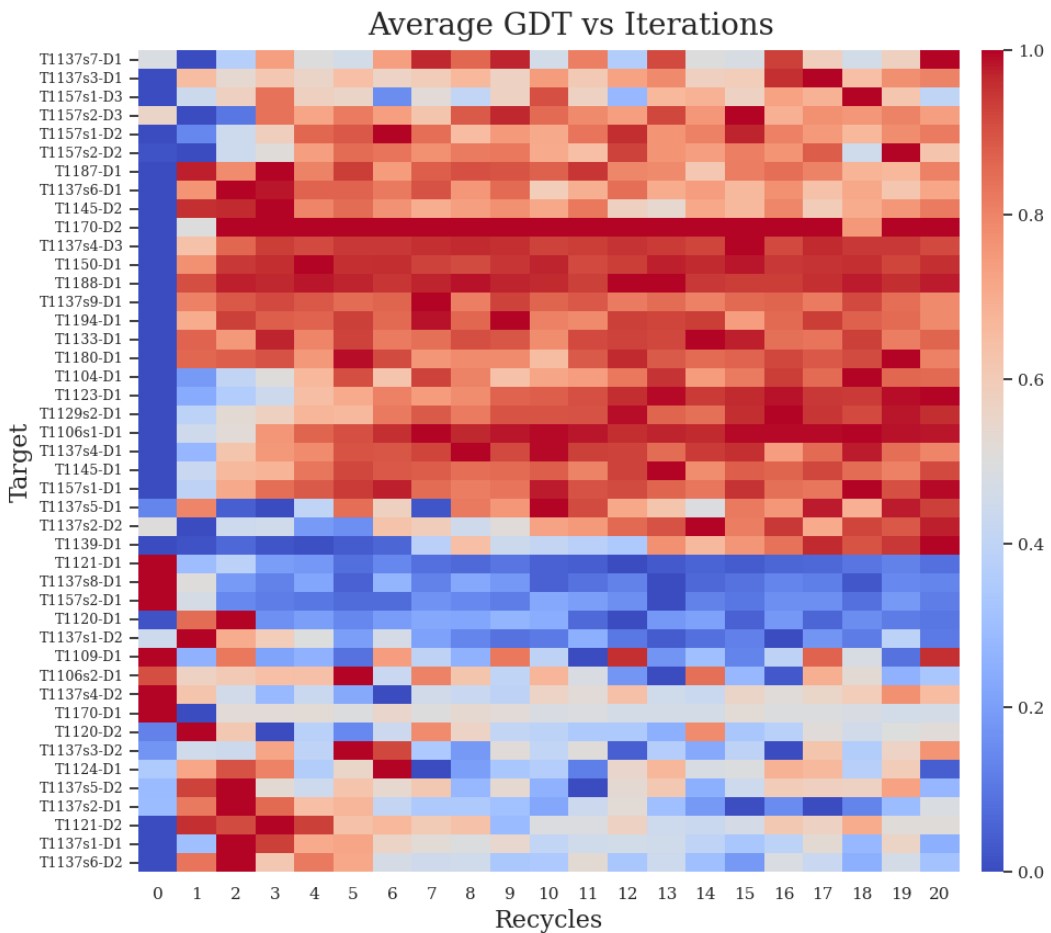

Figure 4: Average GDT scores across all 25 AlphaFold-Multimer predictions (5 seeds per each of the 5 models) for each CASP15 target vs number of recycles, min-max normalized. Larger values are better. The model diverges in GDT for some targets indicating that there is no guaranteed convergence with more recycling.

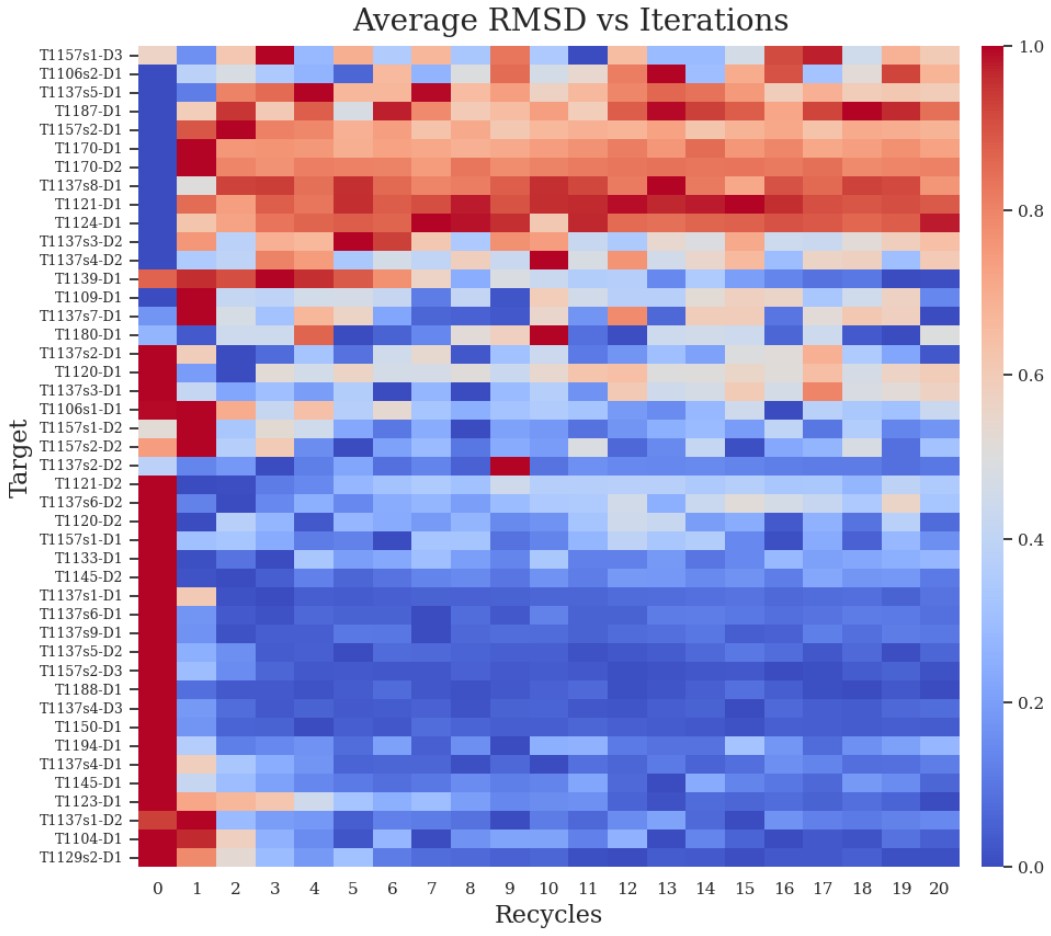

Figure 5: Average RMSD across all 25 AlphaFold-Multimer predictions (5 seeds per each of the 5 models) for each CASP15 target, min-max normalized vs number of recycles. Lower values are better. The model diverges in RMSD for some targets indicating that there is no guaranteed convergence with more recycling.

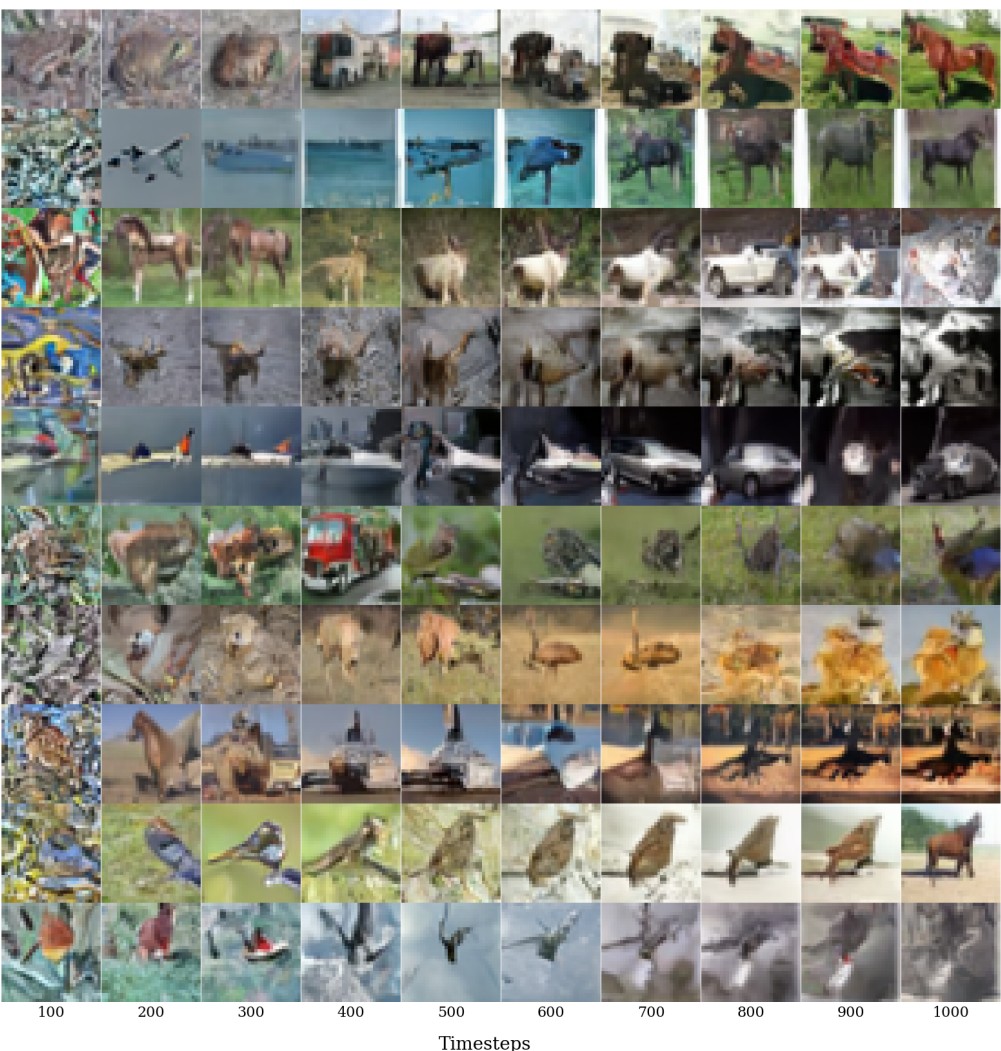

Timesteps

Figure 6: A grid of 10 unconditionally generated images from diffusion models. Each column represents a model trained with a certain number of iterations, and the images are denoised with the same number of iterations. Trained with few iterations, DDPMs are unable to create trajectories back to the data manifold. When trained with at least 400 iterations, the model is able to generate semantically meaningful images.

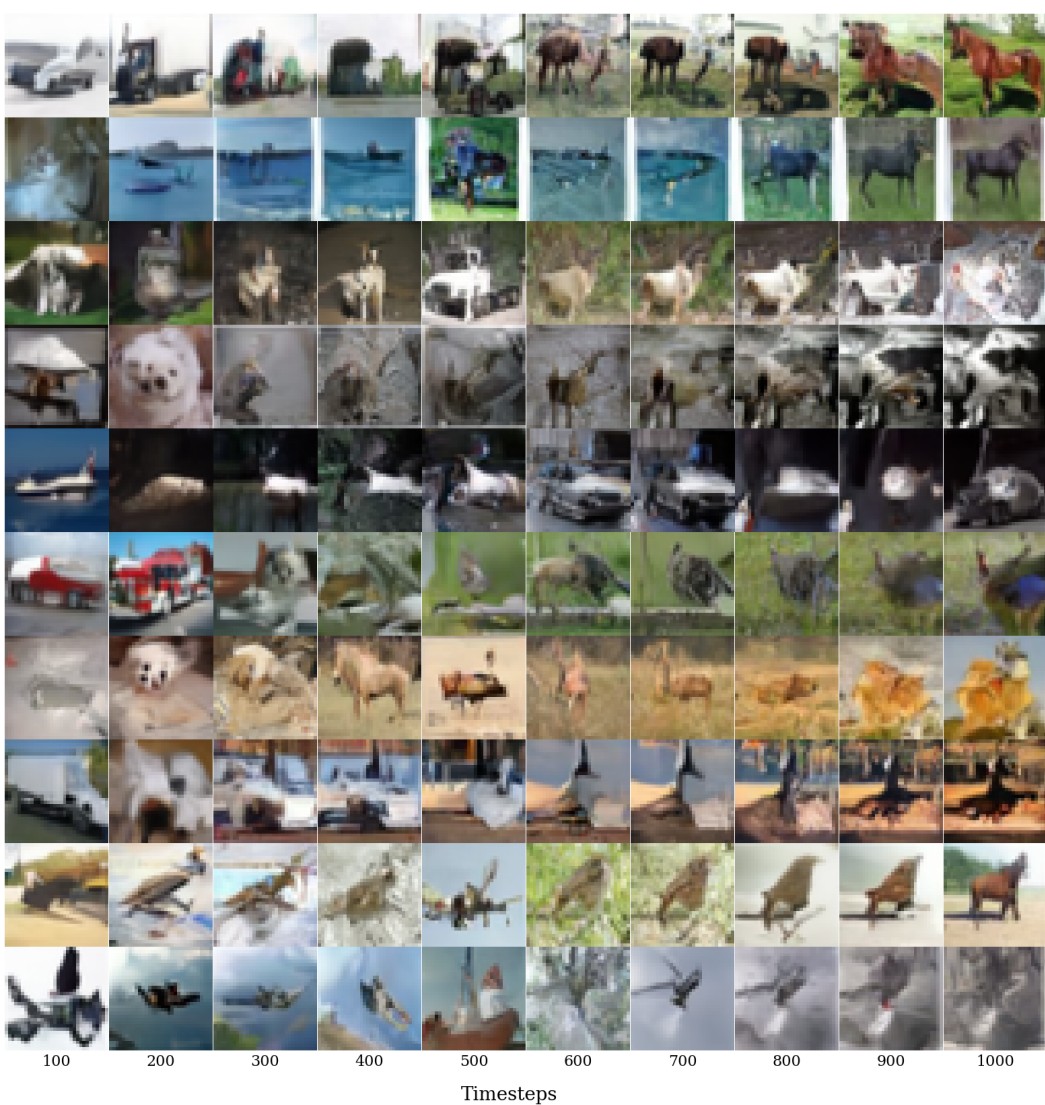

Timesteps

Figure 7: A grid of 10 images where each column represents the number of denoising steps used to return to the data manifold. The same diffusion model trained with 1000 iterations is used for all generations. Note that the same model can produce semantically different images for the same noise vector when the number of timesteps changes, but the reverse trajectories stabilize as the number of timesteps grow.

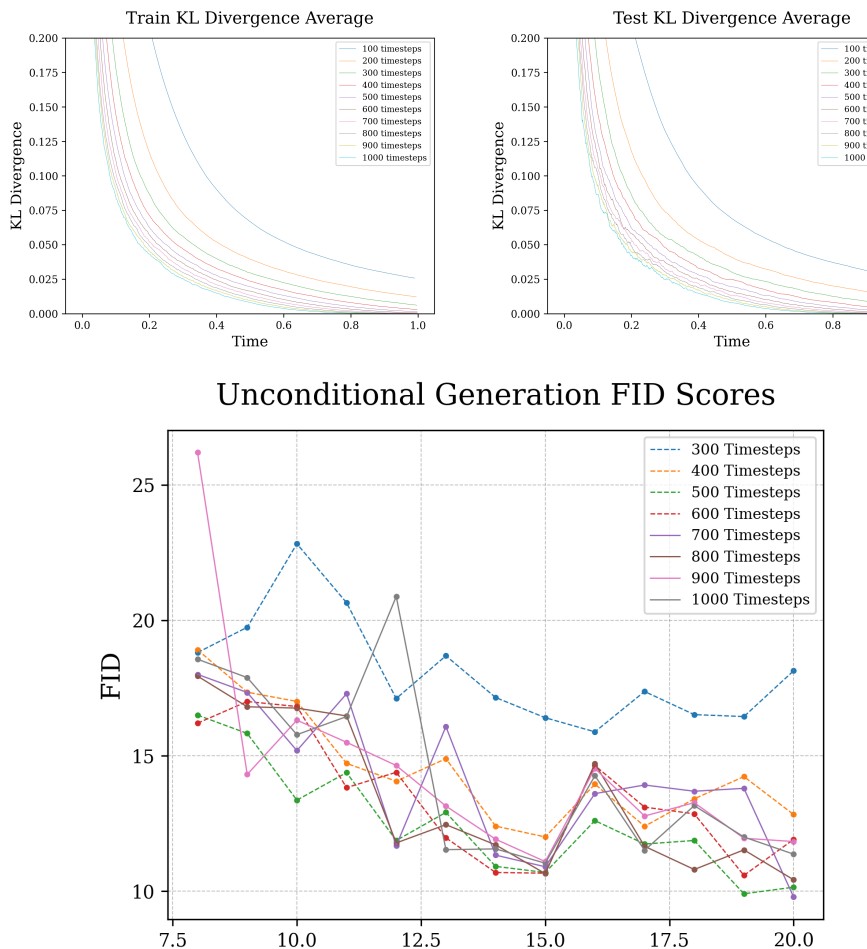

Figure 8: Further analysis with DDPMs. **Top**: The area under the loss curve when predicting the noise is always decreasing with more iterations in DDPMs. Furthermore, the loss curves are roughly monotonically decreasing with time indicating the denoising model converges to a "reasonable" minimum where the model is not better at denoising from a step with more noise compared to one with less noise. The timesteps are normalized to be between 0 and 1. **Bottom**: FID on CIFAR-10 improves with more time steps (in 10000s) at training. After around 500 timesteps, minimal improvement is observed, but FID is never worse. Also, the same amount of training is required to reach optimal FID values regardless of the number of timesteps. All models are UNets with the same exact architecture trained on CIFAR-10's training set with a batch size of 64 images. Convergence on FID was always observed within 150000 steps. Models trained with 100 and 200 timesteps are removed due to their relatively high FID scores, but they also exhibit convergence in FID.

