# OpenReview forum: "Operator Learning Meets Numerical Analysis: Improving Neural Networks through Iterative Methods"
_ICLR.cc/2024/Conference — ICLR 2024 Conference Withdrawn Submission_

### Official Review · Reviewer_yhYn · 2023-10-30

**Soundness:** 2 fair
**Presentation:** 3 good
**Contribution:** 1 poor
**Rating:** 3
**Confidence:** 5

**Summary:**

The authors bridge a link between performing iterative algorithms and network structure on a variety of tasks. They introduce an iterative graph neural network, PIGN, that further demonstrates the benefits of acceleration of the Picard iterations. Overall, the paper provides a new theoretical framework for understanding and improving the performance of deep neural networks. The paper's findings have several implications for future research.  Second, the paper's empirical results suggest that performing iterations through network operators can improve performance.

**Strengths:**

The paper's theoretical framework and empirical results provide new insights into the behavior of deep neural networks, and have the potential to guide the design of new and improved architectures. The authors draw parallels between deep learning and classical numerical analysis, and use fixed point theory to develop better architectures.

The paper demonstrates that performing iterations through network operators improves performance, and introduces an iterative graph neural network, PIGN, that further demonstrates benefits of iterations.

The paper is well-written and well-organized, and the authors clearly articulate their contributions and their implications for future research.

**Weaknesses:**

The perspective that explaining a neural network as an iterative scheme is not new for example [4] and a long line of research starting from [5,6] using optimization schemes to design a neural network. Using a Picard iteration to explain neural structure is not a significant contribution from the reviewer's viewpoint beyond the optimization papers. The structure built in the paper is a special case of the Highway Network [1] and using the momentum in the algorithm to inspire NN structure is also shown in [2,3,4].

Regards the theory (section 3.3) presented by this paper, the reviewer suggests the authors investigate more on deep equilibrium models where theory and experiment may align more.

The idea of the paper is general but the experiment of the paper just runs on several non-standard tasks (for the graph experiments, the networks are not STOA), the VIT experiments are only trained on PDE datasets but not trained on standard CV datasets. What are the criteria to select the test benchmarks in the paper?

[1] Srivastava R K, Greff K, Schmidhuber J. Highway networks. arXiv preprint arXiv:1505.00387, 2015.

[2] Sander M E, Ablin P, Blondel M, et al. Momentum residual neural networks International Conference on Machine Learning. PMLR, 2021: 9276-9287.

[3] Lu Y, Zhong A, Li Q, et al. Beyond finite layer neural networks: Bridging deep architectures and numerical differential equations International Conference on Machine Learning. PMLR, 2018: 3276-3285.

[4] Li H, Yang Y, Chen D, et al. Optimization algorithm inspired deep neural network structure design Asian Conference on Machine Learning. PMLR, 2018: 614-629.

[5] Gregor K, LeCun Y. Learning fast approximations of sparse coding Proceedings of the 27th international conference on international conference on machine learning. 2010: 399-406.

[6] Sun J, Li H, Xu Z. Deep ADMM-Net for compressive sensing MRI[J]. Advances in neural information processing systems, 2016, 29.

**Questions:**

See above

---

### Official Review · Reviewer_S97s · 2023-10-30

**Soundness:** 1 poor
**Presentation:** 1 poor
**Contribution:** 1 poor
**Rating:** 3
**Confidence:** 3

**Summary:**

This paper tries to make a connection between operator learning and iterative methods in numerical analysis.

**Strengths:**

The writing is way too unclear to be able to spot any strength here!

**Weaknesses:**

Page 1 says “Introduces an iterative learning framework for neural networks, supported by theoretical convergence proofs.”. But where exactly is this? This needs to correspond to some theorem or a pseudocode about neural nets and that being shown to be analogous to some theorem on fixed points - and this mapping is simply not apparent anywhere in this paper!

In Section 3, it is simply not clear how any of these general theories of fixed point analysis pertain to the neural architectures that are being targeted to be explained via this approach. Theorem 2 in Section 3 seems to be the main result the authors want to review. But what is the connection between this theorem and anything that is the main result by the authors?

All that one can see in Section 4 are motivational statements and not a concrete mapping.

If the PIGN in Section 5 is the main result, (which it currently looks to be!) then that needs to be defined much earlier than it is occurring just in the experiment section! The description that is given of this is also quite sketchy.

In Section 5.3 its wholly unclear as to what is the loss function or even the new algorithm that is being used to train a ViT to solve the Navier-Stokes.

Also the Lemma 2 in the Appendix is claimed to be about transformers
– but there is no softmax visible there! This is not a transformer at all!

In summary, this submission hardly makes any sense

**Questions:**

Q1.
Are you using the phrase "operator learning" to include things like DeepONets (or any variant of it!)
- which is, to my mind, are most common examples of an operator learning?

Q3.
Can you give a precise mathematical statement (even if stated as conjecture) of what is your claim for the connection between operator learning and Banach-Caccioppoli fixed point theorem that you have stated?

---

### Official Review · Reviewer_uKad · 2023-11-04

**Soundness:** 2 fair
**Presentation:** 2 fair
**Contribution:** 1 poor
**Rating:** 3
**Confidence:** 4

**Summary:**

The paper aims to use classical fixed point methods from numerical analysis to analyze several deep learning algorithms, including diffusion models, AlphaFold, and graph neural networks. After giving a brief introduction to these deep learning frameworks, the authors discuss fix-point methods for solving operator equations (Section 3). They review basic convergence results for a fixed point method based on the Lipschitz constant of the operator $T$ in the target equation. Next, Section 4 has a short discussion on how the fixed-point method could apply to the deep learning frameworks  and Section 5 includes the numerical results of the experiments applying the fixed-point method.

**Strengths:**

1- The fixed-point-based optimization considered in the paper seems interesting and may be used to improve the training of machine learning models.

**Weaknesses:**

1- The paper's theoretical contribution looks marginal to me, and I do not find the paper's analysis related to the theory of deep learning models. The paper essentially proposes using fixed-point-type methods (Equation 3) for training neural networks. The convergence results in Theorem 1 and 2 seem to be simple variations of basic results in numerical analysis and have no specific relation to deep learning algorithms. Also, the fixed-point method in Algorithm 1 can be viewed as applying $L_2$-norm regularization to the learning framework, which is a widely analyzed regularization tool in the literature.

2- It is unclear how the setting discussed in section 3.1 relates specifically to the deep learning frameworks mentioned in the paper. Equation 3.1 states a generic equation format that can represent every equation. I am wondering how analyzing this equation in the general way discussed in the paper could provide any specific understanding of deep learning frameworks.

3- Algorithm 1 (the PIGN method) seems the same as a simple application of gradient descent to optimize an $L_2$-regularized objective function. To see the equivalence, suppose $f$ represents the negative gradient of an objective function $f=-\nabla F$. Then, one can see the application of vanilla gradient descent with stepsize $1-\alpha$ to the objective function $F(x)+\frac{1}{2}\Vert x\Vert^2_2$ is the same as equation 5 in the algorithm. Therefore, Algorithm 1 can be viewed as simply applying gradient-based optimization to an $L_2$-regularized objective function, which lacks the novelty required for an independent contribution.

**Questions:**

1- What is the difference between the proposed Algorithm 1 and the gradient descent optimization for an L2-regularized objective function (Weakness 3 explained above)?

---

### Official Review · Reviewer_16kx · 2023-11-04

**Soundness:** 2 fair
**Presentation:** 2 fair
**Contribution:** 1 poor
**Rating:** 3
**Confidence:** 3

**Summary:**

This paper addresses the lack of theoretical foundations for deep neural networks by considering a theoretical framework that treats neural networks as operators with fixed points representing desired solutions. Subsequently, the authors utilize fixed point theory to establish the convergence of this theoretical framework.

Additionally, the paper points out that widely used deep learning architectures, including diffusion models, AlphaFold, and GNNs, intrinsically embrace this framework. Empirical assessments show that explicitly performing iterations through networks can improve performance. The authors also introduce an iterative graph neural network (PIGN) and showcase its effectiveness in node classification tasks.

**Strengths:**

This paper addresses a significant problem and strives to enhance our understanding of neural network success. Its central and distinctive perspective revolves around the convergence to networks' fixed points by iteratively employing network operators during training.

**Weaknesses:**

1. The framework introduced in this paper appears to deviate from the practical implementation and training of the neural networks they mentioned. Instead, the paper treats neural networks more like implicit models (e.g., [1]) without adequately referencing related works in implicit models. This similarity raises concerns about the main novelty of the paper's approach.

2. The paper only examines simple cases to demonstrate the implications of its theory. For instance, the analysis of transformers without consideration of any nonlinearity makes the discussion and proof trivial. The proof for the basic case of GNNs seems to closely resemble a well-known result in [2], showing the convergence of node features into the eigenspace. This limitation highly weakens the practical implications.

3. While the paper outlines the conditions necessary for ensuring convergence to fixed points, it falls short in providing practical methods to meet these conditions. In contrast, many studies on implicit models (e.g., [1]) approach similar issues from the same perspective but have offered implementations to guarantee these conditions.

These points highlight concerns regarding the novelty of the paper's approach and the practical implications of their theory. The authors must address these issues to bolster the originality of their research.

[1] Gu, F., Chang, H., Zhu, W., Sojoudi, S., & El Ghaoui, L. (2020). Implicit graph neural networks. Advances in Neural Information Processing Systems, 33, 11984-11995.
[2] Oono, K., & Suzuki, T. (2019). Graph neural networks exponentially lose expressive power for node classification. arXiv preprint arXiv:1905.10947.

**Questions:**

1. The paper should provide a clearer explanation of the similarities and distinctions between their proposed framework and implicit models.
2. The paper would benefit from including comprehensive architectural details of the various models under examination.
3. Is it possible to extend the analysis of transformers to incorporate nonlinearity?

---

### Official Review · Reviewer_iEK8 · 2023-11-09

**Soundness:** 1 poor
**Presentation:** 1 poor
**Contribution:** 1 poor
**Rating:** 1
**Confidence:** 4

**Summary:**

In this paper, the authors consider deep learning as fixed point iterations of operator equations. Some theoretical results on convergence can be obtained under conditions. Some popular neural network models, such as diffusion models, AlphaFold, GNN, Neural Integrals, are analysed to show performance improvement with iterations. Picard Iterative Graph Neural Network is proposed and demonstrated with node classification tasks.

**Strengths:**

None.

**Weaknesses:**

- This paper is very poorly written that I even doubt if it is written by human beings.
  - The figures are messy and in some cases not informative at all.
  - Section 3 is hardly related with deep learning. Known results in textbooks are not needed to be detailedly described and rephrased. Section 3.4 is confusing.
  - The proposed GNN model PIGN should not be introduced in the Experiments section.
- Interpreting deep learning from the perspective of optimal control of dynamic systems is not new. There are many previous works that are not discussed nor compared in the paper.
- There are previous works to explicitly take self-consistent iterations as model structures, such as Deep Equilibrium Models, which are not discussed nor compared in the paper.

**Questions:**

None.